# Indole-3-Carboxaldehyde Alleviates LPS-Induced Intestinal Inflammation by Inhibiting ROS Production and NLRP3 Inflammasome Activation

**DOI:** 10.3390/antiox13091107

**Published:** 2024-09-13

**Authors:** Ji Cao, Qiuyu Bao, Haiping Hao

**Affiliations:** 1State Key Laboratory of Natural Medicines, China Pharmaceutical University, Nanjing 210009, China; 3120010077@stu.cpu.edu.cn; 2Key Laboratory of Drug Metabolism and Pharmacokinetics, School of Pharmacy, China Pharmaceutical University, Nanjing 210009, China

**Keywords:** Indole-3-carboxaldehyde, AhR, ROS, NLRP3 inflammasome, intestinal inflammation

## Abstract

Indole-3-carboxaldehyde (IAld) is a tryptophan (Trp) metabolite derived from gut microbiota, which has a potential protective effect on intestinal inflammatory diseases. Abnormal activation of NOD-like receptor family pyrin domain-containing 3 (NLRP3) inflammasome is an important cause of intestinal inflammation. However, the effect and mechanism of IAld on NLRP3 inflammasome activation remain unclear. Here, we found that IAld inhibited the activation of the NLRP3 inflammasome in intestinal epithelial cells, and effectively prevented intestinal epithelial barrier injury caused by lipopolysaccharide (LPS) stimulation. Mechanistically, we demonstrated that IAld activated the aryl hydrocarbon receptor (AhR), subsequently prevented reactive oxygen species (ROS) production, maintained mitochondrial membrane potential, and blocked the NF-κB/NLRP3 inflammatory pathway in intestinal epithelial cells. Also, the AhR-specific inhibitor CH-223191 effectively blocked the IAld-induced NLRP3 inhibition and intestinal epithelial barrier repairment. In addition, in vivo results showed that IAld prevented pro-inflammatory mediator production and intestinal inflammatory damage in LPS-induced mice, which is related to AhR activation and NLRP3 inflammasome inhibition. Collectively, our study unveiled that IAld is an effective endogenous antioxidant and suggested the AhR as a potential treatment target for NLRP3-induced intestinal inflammatory diseases.

## 1. Introduction

The intestine not only plays a crucial role in the digestion and absorption of nutrients but also serves as the most important immune barrier. It has been reported that intestinal inflammation and gut barrier function damage are the main internal causes of intestinal inflammatory diseases [1,2]. In the animal husbandry industry, intestinal homeostasis is closely related to the overall health of animals. Diseases related to intestinal inflammation induced by exogenous pathogenic microorganisms and other factors can seriously affect the metabolic function of the body and reduce the production performance of livestock and poultry, resulting in huge economic losses [3,4,5]. In terms of human health, the occurrence of intestinal diseases, such as inflammatory bowel disease (IBD), can affect the physical and mental health of the human body and, in severe cases, even endanger life [6,7]. Therefore, exploring effective solutions to maintain intestinal homeostasis is of great significance for ensuring human and animal health.

The NLRP3 inflammasome is an important intracellular pattern recognition receptor that can recognize various pathogen-associated pattern molecules (PAMPs) or damage-associated pattern molecules (DAMPs), and is considered to be at the core of regulating inflammatory response [8]. Recent studies have shown that NLRP3 knockout has a protective effect on dextran sulfate sodium (DSS)-induced colitis mice [9], and NLRP3 inhibitors can effectively improve the occurrence of colitis [10]. Excessive production of pro-inflammatory factors mediated by the NLRP3 inflammasome can damage the intestinal epithelial barrier function, leading to intestinal inflammation [11,12]. In the piglet enteritis model, the intestinal epithelium NLRP3 inflammasome is abnormally activated, which can lead to a large release of pro-inflammatory cytokines such as interleukin-1β (IL-1β) and interleukin-18 (IL-18) [13]. Our previous research found that the cholesterol metabolism intermediate dehydroepiandrosterone (DHEA) can negatively regulate NLRP3 inflammasome activation and alleviate intestinal inflammation in colitis mice, and may serve as a nutritional supplement for preventing IBD [14]. To sum up, NLRP3 inflammasome activation needs to be strictly controlled to prevent the occurrence of intestinal inflammation.

Tryptophan (Trp) is an essential amino acid that can only be obtained from dietary sources, and it can be metabolized and converted by the gut microbiota into an endogenous ligand for the aryl hydrocarbon receptor (AhR) [15]. As an important component of the immune barrier, the AhR can promote epithelial cell recovery and maintain intestinal barrier integrity [16,17]. In colitis mice, pharmacological activation of the AhR can improve the pathological symptoms of intestinal inflammation [18], while AhR knockout increases susceptibility to intestinal inflammation [19]. Among the numerous AhR ligands, indole-3-aldehyde (IAld) can effectively enhance intestinal epithelial barrier function and inhibit the occurrence of colitis in mice, suggesting that IAld can maintain intestinal homeostasis [20,21,22]. However, the effect and mechanism of IAld (an AhR ligand) on the regulation of NLRP3-induced inflammatory response are still unclear.

In the present study, lipopolysaccharide (LPS)-induced intestinal inflammation models (in vivo and in vitro) were used to explore the molecular mechanism of microbial-derived IAld in the regulation of NLRP3 inflammasome activation and gut barrier function. We aim to explore a new strategy for alleviating intestinal inflammatory diseases from the perspective of regulating host function by the gut microbiota-derived metabolites.

## 2. Materials and Methods

### 2.1. Reagents and Antibodies

Indole-3-carboxaldehyde (IAld) and lipopolysaccharide (LPS, *Escherichia coli* 055: B5) were obtained from Sigma–Aldrich (St Louis, MO, USA). The fetal bovine serum (FBS) and Dulbecco’s modified Eagle’s medium (DMEM) were obtained from Gibco (Erie, NY, USA). The AhR inhibitor CH-223191 was provided by MCE (St. Louis, MO, USA). The rabbit anti-ZO-1 (BS71522), occludin (BS72035), NLRP3 (BS90948), NF-κB p65 (BS1257), IL-1β (BS65005), Caspase-1 (BS65650), β-Tubulin (AP0064), GAPDH (AP0063), and HRP goat anti-rabbit IgG (BS13278) antibodies were purchased from Bioword (Nanjing, China). The rabbit anti-AhR (A4000) antibody was purchased from ABclonal (Wuhan, China). The rabbit anti-iNOS (22226-1-AP) antibody was provided by ProteintechGroup (Rosemont, IL, USA). The goat anti-rabbit IgG (H + L) FITC (GB22303) and goat anti-rabbit IgG (H + L) Cy3 (GB21303) antibodies were purchased from Servicebio (Nanjing, China).

### 2.2. Intestinal Epithelial Cell Culture and LPS Treatment

The pig intestinal epithelial cell line IPEC-J2 was kindly donated by prof. Yuanshu Zhang and Dr. Zhiqiang Li from Nanjing Agricultural University. The human colon epithelial cell line Caco2 was kindly donated by prof. Haitian Ma from Nanjing Agricultural University. IPEC-J2 and the Caco2 cells were grown in the Dulbecco’s modified Eagle’s medium (DMEM) containing 10% fetal calf serum (FBS, Gibco, New York, NY, USA) and 1% penicillin-streptomycin (Gibco, New York, NY, USA) at 37 °C in a 95% humidified atmosphere with 5% CO_2_. The cells were passaged every 2 days. Referring to previous studies [14,23], the intestinal epithelial cells (IPEC-J2 and Caco2 cells) were exposed to LPS (10 μg/mL) for 24 h to establish the inflammatory injury model in vitro.

### 2.3. Animals and Treatments

The 6–8 week-old male C57BL/6 mice weighing 18 to 20 g (purchased from Beijing Vital River Laboratory Animal Technology Co., Ltd. Beijing, China) were injected intraperitoneally with LPS (20 mg/kg body weight) to induce an inflammatory model. All mice were given adaptive feeding for 7 days before the experiment. Additionally, the mice were housed in a specific pathogen-free (SPF) grade room with a constant room temperature (24 ± 1 °C) and were allowed to water and feed ad libitum. The mice were randomly divided into 3 groups: normal control (CTL) (*n* = 3), LPS (20 mg/kg body weight) treatment (*n* = 4), LPS (20 mg/kg body weight) plus IAld (50 mg/kg body weight) treatment (*n* = 5). The dose of IAld used in the present study was selected based on the literature previously published by ourselves and others [24,25,26]. The dose of LPS used in the present study was also selected based on the literature previously published by ourselves and others [27,28]. The IAld oral treatment (once daily gavage) was started 5 days before LPS stimulation (intraperitoneal injection), the other mice received an equal amount of vehicle (0.5% carboxymethylcellulose (CMC)). The mice were sacrificed after 12 h or 24 h of LPS stimulation, and the livers, kidneys, lungs, colons, and serum samples were collected for further ex vivo detection. The specimens of the lung and colon were fixed with 10% formalin for H&E staining. All animal experiments were conducted in strict accordance with protocols approved by the Animal Welfare & Ethical Committee of China Pharmaceutical University.

### 2.4. Immunofluorescence (IF)

The IF analysis was performed as previously described [14]. In brief, the fixed colon tissues or fixed cells were incubated with the indicated primary antibody (dilution ratio of 1:100), followed by incubation with the fluorescently labeled secondary antibodies (dilution ratio of 1:500). The cells were dyed with DAPI and analyzed by the laser scanning confocal microscope (Zeiss LSM 700, Jena, Germany).

### 2.5. DCFH-DA, MitoTracker Red, and JC-1 Staining

The production of intracellular ROS and mitochondrial membrane potential were detected using the DCFH-DA (Beyotime, Shanghai, China), Mito-Tracker Red CMXRos (Beyotime, Shanghai, China), and JC-1 fluorescent probe (Servicebio, Nanjing, China) following the manufacturer’s instructions, respectively. Briefly, the cells were added with DCFH-DA (10 μM), Mito-Tracker Red CMXRos (100 nM), and JC-1 fluorescent probe for 20–30 min. After incubation and washing with phosphate-buffered solution (PBS, pH = 7.4), the fluorescence intensity was immediately detected using a Zeiss LSM 710 (Jena, Germany) confocal microscope or an ordinary inverted fluorescence microscope (Olympus, Tokyo, Japan).

### 2.6. Intestinal Epithelial Cell Permeability Analysis

The intestinal epithelial cell permeability analysis was performed as previously described [14]. In brief, IPEC-J2 and Caco2 cells were seeded in the upper chambers of 24-well Transwell^®^ plates (0.4 μm pore polyester membranes; Biofil, Guangzhou, China) until a stable monolayer was formed. After the indicated treatments, FITC-Dextran (FD-4, MW 4000; Sigma Aldrich, St Louis, MO, USA) solution (1 mg/mL, 0.2 mL) was added to the upper chambers. After 24 h of incubation, the samples were collected from the bottom chamber, and the fluorescence 96-well plate reader (excitation wavelength 480 nm, emission wavelength 525 nm) was used to detect the FD-4 flux.

### 2.7. Western Blotting (WB)

Total proteins of cells or colon tissues were extracted with RIPA lysis buffer (Servicebio, Nanjing, China) as per the manufacturer’s protocol. The concentrations of proteins were determined by the Pierce BCA protein assay kit (Beyotime, Shanghai, China). Western blotting was applied to detect the indicated protein expression levels as previously described [29]. Briefly, the extracted proteins were separated by SDS-PAGE on 10% gel and transferred into polyvinylidene fluoride (PVDF) membranes (0.45 μm; GE Healthcare, Madison, WI, USA). The membranes were blocked with Tris-buffered saline Tween (TBST) containing 5% non-fat milk for 1 h, then washed the membranes three times with TBST and incubated overnight at 4 °C with indicated rabbit polyclonal antibodies (dilution ratio of 1:1000). After washing five times (5–10 min per time) with TBST, the membranes were incubated with HRP goat anti-rabbit IgG (dilution ratio of 1:10,000) for another 1 h at 37 °C. The protein bands were analyzed by using a clarity ECL chemiluminescent substrate (purchased from Vazyme Biotech Co., Ltd., Nanjing, China), and the ImageJ software (Version 1.48) was used to quantify indicated blots. The GAPDH (dilution ratio of 1:10,000) or β-Tubulin (dilution ratio of 1:10,000) was used as a control, and the results were presented as the fold changes in target protein expression normalized to the control.

### 2.8. Real-Time Quantitative PCR (qPCR)

The total RNA of the liver and kidney samples was extracted using Trizol reagent according to the manufacturer’s instructions (Vazyme, Nanjing, China), and the total RNA was reverse transcribed into cDNA by the reverse transcriptase kits (Vazyme, Nanjing, China). The methods for the qPCR and the primers used in the present study were performed as previously described [30]. In brief, the samples were analyzed by using the ABI 7500 Real-time Detection System (Applied Biosystems, USA) and programmed to conduct one cycle (95 °C for 3 min) and 40 cycles (95 °C for 10 s, 60 °C for 30 s, and 72 °C for 20 s). The sequences of primers used in the present study were shown as follows: F 5′-CCTTGTTCAGCTACGCCTTC-3′, R 5′-CTGAGGGCTCTGTTGAGGTC-3′ (for the *iNOS* gene); F 5′-CTACCGTCGTGACTTCGC-3′, and R 5′-GGGTGACATCTCCCTGTT-3′ (for the *β-actin* gene). The 2^−△△CT^ method was used to calculate the relative *iNOS* gene expression level, and the *β-actin* was used as the internal reference gene.

### 2.9. Enzyme-Linked Immuno-Sorbent Assay (ELISA)

The TNF-α, IL-1β and IL-6 concentrations in serum were detected using the ELISA kits (ExCell Bio) according to the manufacturer’s instructions.

### 2.10. Statistical Analysis

All data were presented as mean ± SEM and the results of at least three independent experiments. One-way analysis of variance (ANOVA) and unpaired Student’s t-test (two-tailed) were performed by GraphPad Prism 8.0.2 software to compare the significant differences among different treatments. Student’s t-test was used to compare the differences between two groups, and one-way ANOVA was used for multiple group comparisons (data following a normal distribution). The differences were considered significant at *p* < 0.05 and extremely significant at *p* < 0.01.

## 3. Results

### 3.1. IAld inhibits LPS-Induced NLRP3 Inflammasome Activation in Intestinal Epithelial Cells

The NLRP3 inflammasome is the core cause of inflammation and is involved in the occurrence of intestinal inflammatory diseases [31,32]. LPS is one of the important virulence factors of Gram-negative bacteria, which can cause excessive accumulation of ROS, activation of NLRP3 inflammasome, and barrier dysfunction in intestinal epithelial cells [33,34,35]. Thus, we established an inflammatory model of intestinal epithelial cells IPEC-J2 by using LPS stimulation. As shown in Figure 1A,B, the confocal imaging confirmed that 10 μM of IAld treatment (at physiological concentration level) can prevent LPS-induced upregulation of NLRP3 protein expression level. The Western blots analysis confirmed that LPS can upregulate the expression level of the NLRP3, caspase-1, and IL-1β proteins (indicating the activation of the NLRP3 inflammasome), while IAld treatment can downregulate the expression level of NLRP3-related proteins in a dose-dependent manner (Figure 1C). We further detected ROS levels in cells through DCFH-DA staining, as excessive accumulation of ROS is the key upstream time for inducing abnormal activation of the NLRP3 inflammasome [36,37]. LPS stimulation led to a large amount of ROS production in intestinal epithelial cells, while IAld almost completely prevented the excessive generation of ROS in IPEC-J2 cells (Figure 1D,E).

We validated the above findings by using another type of intestinal epithelial cell line Caco2 (human source). As shown in Figure 2A, the confocal imaging confirmed that IAld can prevent LPS-induced upregulation of NLRP3 protein expression level. Western blot analysis confirmed this result and found that different concentrations of IAld treatment (especially 10 μM and 50 μM) can significantly reduce the caspase-1 and IL-1β protein expression levels (Figure 2B). In addition, IAld effectively prevented LPS-induced excessive production of ROS in Caco2 cells (Figure 2C). Together, these results illustrate that IAld can inhibit LPS-induced ROS production and NLRP3 inflammasome activation in intestinal epithelial cells.

### 3.2. IAld Prevents LPS-Induced Mitochondrial Dysfunction in Intestinal Epithelial Cells

Excessive accumulation of ROS can cause mitochondrial dysfunction and exacerbate the inflammatory response by activating nuclear factor kappa B (NF-κB) and the NLRP3 inflammasome signaling pathway. Therefore, we examined the effect of IAld on LPS-mediated mitochondrial dysfunction in intestinal epithelial cells. As shown in Figure 3A–C, LPS treatment significantly upregulated the level of JC-1 monomers in IPEC-J2 cells, indicating a downregulation of mitochondrial membrane potential. The MitoTracker Red staining results further confirmed LPS-induced mitochondrial dysfunction (Figure 3D,E). It is worth noting that IAld treatment dramatically improved the mitochondrial function in LPS-stimulated IPEC-J2 cells, which is reflected in the increase in relative JC-1 aggregate level and MitoTracker Red fluorescence intensity (Figure 3A–E). We also found that IAld can alleviate the mitochondrial damage in LPS-induced Caco2 cells (Figure 3F,G). These results further demonstrated that IAld can inhibit NLRP3 activation by preventing ROS-induced mitochondrial functional damage in intestinal epithelial cells.

### 3.3. IAld Alleviates LPS-Induced Barrier Dysfunction in Intestinal Epithelial Cells

Intestinal inflammation is often accompanied by damage to the intestinal barrier function (i.e., increased intestinal permeability), which is mainly determined by tight junctions (TJs). A TJ, as a protein complex, mainly includes Zonula occludens (ZO) and occludin proteins. Therefore, we further investigated the effect of IAld on intestinal epithelial barrier function. As shown in Figure 4A,B, the confocal imaging indicated that IAld can prevent LPS-induced damage to the distribution of tight junction protein ZO-1. In addition, 10 μM of IAld treatment significantly increased the expression level of ZO-1 protein in IPEC-J2 cells (Figure 4C). In Caco2 cells, IAld treatment also improved the damage to ZO-1 distribution caused by LPS (Figure 4E,F). Moreover, IAld (50 μM and 100 μM) treatments significantly increased the protein expression levels of ZO-1 and occludin in LPS-stimulated cells (Figure 4G).

We also tested intestinal epithelial permeability in vitro based on the Transwell cell culture. As shown in Figure 4D,H, LPS treatment significantly increased FD-4 leakage from the upper chambers, while IAld treatment improved the intestinal permeability both in IPEC-J2 and Caco2 cells, which is consistent with the enhancement of TJ barrier function. These above results suggested that IAld alleviates LPS-induced barrier dysfunction in intestinal epithelial cells, which is closely related to its ability to alleviate inflammatory response.

### 3.4. IAld Alleviates Intestinal Inflammatory Injury in an AhR-Dependent Manner

Subsequently, we hope to explore the potential mechanisms by which IAld inhibits NLRP3 inflammasome activation and intestinal barrier dysfunction. It has been shown that IAld is an endogenous ligand of the AhR [22], so we speculate that IAld may exert anti-inflammatory function by activating the AhR. As shown in Figure 5A, IAld significantly increased the AhR protein expression level in LPS-stimulated Caco2 cells, while AhR-specific inhibitor CH-223191 significantly downregulated the expression level of AhR protein (indicating that the AhR signal has been successfully suppressed). In addition, the DCFH-DA staining results revealed that CH-223191 prevented IAld from reducing LPS-induced ROS over-production (Figure 5B,C).

As excessive production of ROS can cause activation of the NF-κB/NLRP3 signaling pathway [38], we further investigated the impact of CH-223191 plus IAld treatment on NF-κB/NLRP3 activation. The confocal imaging confirmed that LPS significantly enhanced the expression of NF-κB activation key factor p65 protein, while IAld treatment significantly reduced the p65 levels, which is associated with AhR activation (Figure 5D). Western blot analysis confirmed that the effect of IAld on reducing IL-1β protein levels is blocked by CH-223191 in LPS-stimulated Caco2 cells (Figure 5E). Importantly, AhR inhibition prevented the intestinal barrier protective effect of IAld, which was demonstrated by ZO-1 immunofluorescence detection (Figure 5F) and the FD-4 leakage experiments (Figure 5G). Importantly, CH-223191 treatment weakened the protective effect of IAld against LPS-induced mitochondrial membrane potential damage (Figure 5H,I). In IPEC-J2 cells, Western blot analysis confirmed the inhibitory effect of CH-223191 on AhR signaling (Figure 6A). Notably, AhR inhibition impairs IAld’s ability to alleviate LPS-induced ROS over-production (Figure 6B), NF-κB activation (Figure 6C), and TJ barrier dysfunction (Figure 6D,E). The above results indicated that the effect of IAld in alleviating intestinal epithelial cell inflammatory response (the NF-κB/NLRP3 activation) and gut barrier damage was at least partially related to AhR activation.

### 3.5. IAld Alleviates LPS-Induced Intestinal Inflammatory Response in Mice

As shown in Figure 7A, LPS intraperitoneal injection caused severe tissue damage to the lungs and colons of mice (such as inflammatory infiltration and loss of intestinal crypts). IAld treatment ameliorated LPS-induced pathological damage to the lungs. Besides, the IAld treatment group had a more complete morphology of the colon tissue (Figure 7A). The ELISA results showed that IAld obviously reduced the pro-inflammatory cytokine TNF-α in the serum of LPS-stimulated mice, and downregulated the pro-inflammatory cytokine (IL-1β, IL-6) concentration in the serum to some extent (Figure 7C–E). Inducible nitric oxide synthase (iNOS) is a classic inflammatory factor that can be stimulated by LPS or pro-inflammatory cytokines (e.g., TNF-α). As shown in Figure 7B,F, the iNOS transcription levels were markedly elevated in the liver and spleen of LPS-stimulated mice. IAld treatment reduced the transcription level of iNOS in the liver to a certain extent (Figure 7B) and significantly decreased the iNOS transcription level in the kidney (Figure 7F). In the colons of mice, we confirmed that IAld can inhibit LPS-induced expression of iNOS protein levels through immunofluorescence results (Figure 7G). In addition, LPS stimulation significantly reduced the expression level of the ZO-1 protein in the colon (indicating intestinal barrier dysfunction), while IAld significantly increased the expression level of the ZO-1 protein (Figure 7H), which confirms the role of IAld in alleviating inflammatory response and intestinal barrier function in LPS-stimulated mice.

We further confirmed the role of the AhR and NLRP3 inflammasomes in LPS-induced intestinal inflammation. As shown in Figure 7I,J, LPS significantly increased the expression level of the caspase-1 protein in the colon of mice (indicating the activation of NLRP3 inflammasomes) and decreased the expression level of AhR proteins, while IAld treatment inhibited NLRP3 inflammasome activation and increased AhR protein expression level to some extent. Together, the above data demonstrated that IAld can effectively prevent intestinal inflammation and barrier damage caused by LPS, which is closely associated with AhR activation and ROS-NLRP3 signal inhibition (Figure 7K).

## 4. Discussion

Metabolites derived from gut microbiota are key signaling molecules that mediate the interaction between gut microbiota and the host, playing an important role in regulating the body’s immune and inflammatory responses [39]. Recently, supplementation therapy based on gut microbiota metabolites has been considered as an effective new strategy for preventing and treating intestinal inflammatory diseases [40]. In this study, we mainly focused on the regulatory effect and mechanism of the microbiota-derived tryptophan metabolite IAld on the intestinal inflammatory response (ROS-induced NLRP3 inflammasome activation). We found that IAld activated AhR signaling, subsequently inhibited ROS production, and blocked the NF-κB/NLRP3 pathway in intestinal epithelial cells, which is necessary for the maintenance of intestinal barrier function. Based on the above results, we concluded that IAld can negatively regulate the NLRP3 inflammasome and serve as a promising antioxidant for the treatment of intestinal inflammation-related diseases.

The NLRP3 inflammasome is the core of inflammatory response and participates in the occurrence and development of intestinal inflammatory diseases [41]. Therefore, targeted regulation of NLRP3 activation will be a promising strategy for preventing and treating inflammation-related diseases [31]. However, there are currently no FDA-approved NLRP3 inhibitors being used in clinical practice. The gut microbiota can produce various metabolites, such as short-chain fatty acids (SCFAs), bile acid, and indole derivatives to regulate intestinal homeostasis. Xiao N. et al.’s study found that egg yolk lipids can alleviate DSS-induced colitis in mice by increasing the production of SCFAs and then inhibiting NLRP3 inflammasome activation [42]. In house dust mite (HDM)-induced neutrophilic asthma mice, the three main SCFAs can mitigate lung inflammation by suppressing the NLRP3 signal [43]. Bile acids also play an important regulatory role in NLRP3 inflammasome-induced inflammatory responses. For example, gut microbiota-mediated secondary bile acid (deoxycholic acid (DCA)) can alleviate Staphylococcus aureus-induced mastitis by inhibiting NLRP3 inflammasome activation [44]. In addition, Indoles (including indole-3-acetic acid and indole-3-propionic acid) have been reported to prevent the assembly activation of the NLRP3 inflammasome, reduce the production of inflammatory factors, and ultimately inhibit neuroinflammation in APP/PS1 mice [45]. The above studies indicate that metabolites derived from gut microbiota could play a key role in regulating NLRP3 inflammasome activation. However, the regulatory effect of microbiota-derived tryptophan metabolite IAld on the NLRP3 inflammasome is currently unclear. In our study, based on the known anti-inflammation effects of IAld [46], and the notion that the NLRP3 inflammasome plays a key role in the occurrence of intestinal inflammatory diseases [47], we further discovered the effect of IAld on NLRP3 inflammasome activation in intestinal epithelial cells. Our results indicate that IAld can negatively regulate NLRP3 inflammasome activation, inhibit the production of pro-inflammatory factors, and ultimately prevent intestinal epithelial barrier damage in LPS-stimulated intestinal epithelial cells. More importantly, we have demonstrated in animal experiments that the protective effect of IAld on intestinal inflammation is associated with the NLRP3 inflammasome, as IAld treatment significantly reduced the LPS-induced expression level of the caspase-1 protein, an NLRP3 inflammasome effector. Nuclear factor kappa B (NF-κB) is the upstream signaling pathway of the NLRP3 inflammasome, which is involved in the production of pro-inflammatory factors (such as iNOS) [48,49]. Our in vitro experimental results also confirmed the inhibitory effect of IAld on NF-κB signaling. In addition, IAld reduced the upregulation of iNOS protein levels in LPS-induced intestinal tissue of mice and prevented the release of pro-inflammatory cytokines (especially the content of TNF-α in serum). TNF-α can damage the intestinal barrier function and exacerbate intestinal inflammatory response, and anti-TNF therapies are of great significance for the prevention and control of intestinal inflammatory diseases, such as inflammatory bowel disease (IBD) [50]. The decrease in TNF-α levels may partially explain the anti-inflammatory effect of IAld. These above results indicated that IAld can inhibit the NF-κB/NLRP3 pathway, thereby alleviating LPS-induced inflammation in intestinal epithelial cells.

Mitochondria are the center of cellular respiration and energy metabolism, and their respiratory chain function depends on the balance between the pro-oxidative and antioxidant systems. Mitochondrial respiration drives the production of adenosine triphosphate (ATP) through electron transfer and the proton gradient, with reactive oxygen species (ROS) being a natural byproduct of this process [51]. Bacterial-derived lipopolysaccharides (LPSs) can stimulate mitochondria in cells to produce a large amount of ROS, while excessive levels of ROS can cause damage to proteins, lipids, and DNA, leading to mitochondrial dysfunction and exacerbating the body’s inflammatory response by activating NF-κB/NLRP3 signaling [52,53,54]. Even worse, excessive activation of the NF-κB/NLRP3 signal can disrupt intestinal epithelial barrier function, leading to further amplification of intestinal inflammation [55]. Therefore, targeted maintenance of mitochondrial function contributes to the inhibition of inflammatory responses in gastrointestinal diseases [56]. Gut microbiota signaling (including small molecule metabolites derived from gut microbiota) to mitochondria has been shown to alter mitochondrial function, regulate inflammasome signaling, and alter the epithelial barrier function [57]. In addition, the interaction between mitochondria and gut microbiota is crucial for gut health [58]. Ge et al. found that vitamin B12 (VB12) can regulate ileal epithelial mitochondrial homeostasis and gut microbiota composition to resist Salmonella infection in mice [59]. Our recent research suggests that another microbiota-derived tryptophan metabolite indole-3-acetic acid (IAA) can interfere with the mitochondrial bioenergetics of intestinal stem cells, leading to hindered differentiation of secretory cell lineages and disrupt intestinal homeostasis [60]. In the present study, we found that IAld significantly improved LPS-induced mitochondrial dysfunction (manifested as a decrease in mitochondrial membrane potential) in the intestinal epithelial cells. Correspondingly, IAld effectively prevented LPS-induced intracellular ROS over-production, NLRP3 inflammasome activation, and intestinal mucosal barrier injury. Our results indicated that the regulatory effect of IAld on ROS-NLRP3 signaling depends on the maintenance of mitochondrial function.

The AhR is a key transcription factor for regulating intestinal homeostasis [61]. In macrophages, AhR activation can inhibit NLRP3 transcription and prevent NLRP3 inflammasome activation [62]. In non-alcoholic steatohepatitis (NASH) mice, paeonol can increase the levels of indole-3-acetic acid, subsequently activating the AhR and inhibiting NLRP3 inflammasome activation [63]. In DSS-induced colitis mice, indole-3-acetic acid can also regulate the AhR/NRF2/NLRP3 pathway to alleviate intestinal inflammation and barrier function damage [64]. In addition, naringenin has been reported to activate the AhR signal to prevent NLRP3 activation, ultimately protecting against acute pancreatitis-associated intestinal injury [65]. The above studies indicate that AhR/NLRP3 signaling can be a potential therapeutic target for inflammation-related diseases. It is worth noting that the function of the AhR seems to be related to the type of its ligand, as benzo (a) pyrene (BaP) can activate AhR/ERK signaling, thereby increasing ROS production [66]. Additionally, the rubber anti-aging agent N-(1,3-Dimethylbutyl)-N’-phenyl-phenylenediamine quinone (6PPDQ) can also promote ROS production through AhR, leading to the interruption of autophagy flux and ultimately causing cardiomyocyte senescence. IAld has been reported to regulate AhR/IL-22 signaling to regulate intestinal barrier repair, thereby alleviating intestinal inflammation [21,67]. However, the role of IAld in NLRP3-mediated intestinal inflammation is still unclear, and the mechanism of IAld on NLRP3 inflammasome activation in intestinal epithelial cells needs to be explored. In the present study, we found that the endogenous AhR ligand IAld can play a role in preventing mitochondrial damage and ROS production in intestinal epithelial cells, ultimately preventing NLRP3 inflammasome activation. In the LPS-stimulated mouse intestinal inflammation model, IAld treatment can also activate AhR signaling and inhibit downstream inflammatory pathway activation. Further indicating that IAld may serve as a beneficial supplement with significant importance in maintaining human and animal intestinal health.

Our current research also has some limitations. Firstly, this study only investigated the regulatory effect of IAld, one microbial-derived tryptophan metabolite, on the ROS-NLRP3 signaling pathway and its protective mechanism against intestinal inflammation. There are many other microbe-derived tryptophan metabolites in the body, such as indole-3-lactic acid and indole-3-propionic acid, which are also important ligands of the AhR. In future research, further exploration of the regulatory effects of other AhR ligands on the ROS-NLRP3 signaling pathway in intestinal inflammation is needed. Secondly, although the Caco2 and IPEC-J2 cell lines were used as in vitro models in the present study, these cell lines cannot fully simulate the physiological state of the gut in vivo. Future research can use intestinal organoid culture techniques to validate the biological functions of IAld, as intestinal organoids have physiological structures and function in a manner that is similar to the real intestine. Thirdly, the use of AhR knockout mice will help clarify whether the regulatory effect of IAld on NLRP3 inflammasome activation depends on the AhR in vivo. In addition, it is necessary to further use other representative animal intestinal inflammation models and increase the sample size of animal experiments. Last but not least, we cannot rule out the possibility that the antioxidant and anti-inflammatory effects of IAld may be mediated through AhR-independent mechanisms, such as regulating gut microbiota composition or activating other receptors. A recent study suggests that three tryptophan-derived metabolites (indole-3-ethanol (IEt), indole-3-pyruvate (IPyA), and indole-3-aldehyde (I3A)) can activate dopamine receptor D2 (DRD2) to confer colonization resistance [68]. Therefore, further investigation is needed to elucidate the mechanism by which microbial-derived tryptophan metabolites regulate host function.

## 5. Conclusions

In conclusion, our data indicated that gut microbiota-derived IAld alleviates LPS-induced mitochondrial dysfunction, excessive production of ROS, and inflammatory damage in the intestinal epithelium by activating the AhR and inhibiting the NLRP3 inflammasome both in vivo and in vitro. These findings suggested that the AhR ligand IAld is an endogenous NLRP3 inhibitor that may serve as a potential agent for treating intestinal inflammation. Given the important role of the NLRP3 inflammasome in inflammatory responses, we also have reason to infer the therapeutic prospects of IAld for other NLRP3-related inflammatory diseases such as Alzheimer’s disease (AD), nonalcoholic fatty liver disease (NAFLD), and multiple sclerosis (MS). In future clinical applications, in addition to directly supplementing IAld, we can also target and regulate the abundance of IAld-producing probiotics (e.g., *Lactobacillus reuteri*) in the gut microbiota. Furthermore, we need to further investigate the regulatory effects of other AhR ligands on ROS-induced NLRP3 inflammasome activation, which will help reveal the molecular mechanism by which gut microbiota regulates the host antioxidant states.

## Figures and Tables

**Figure 1 antioxidants-13-01107-f001:**
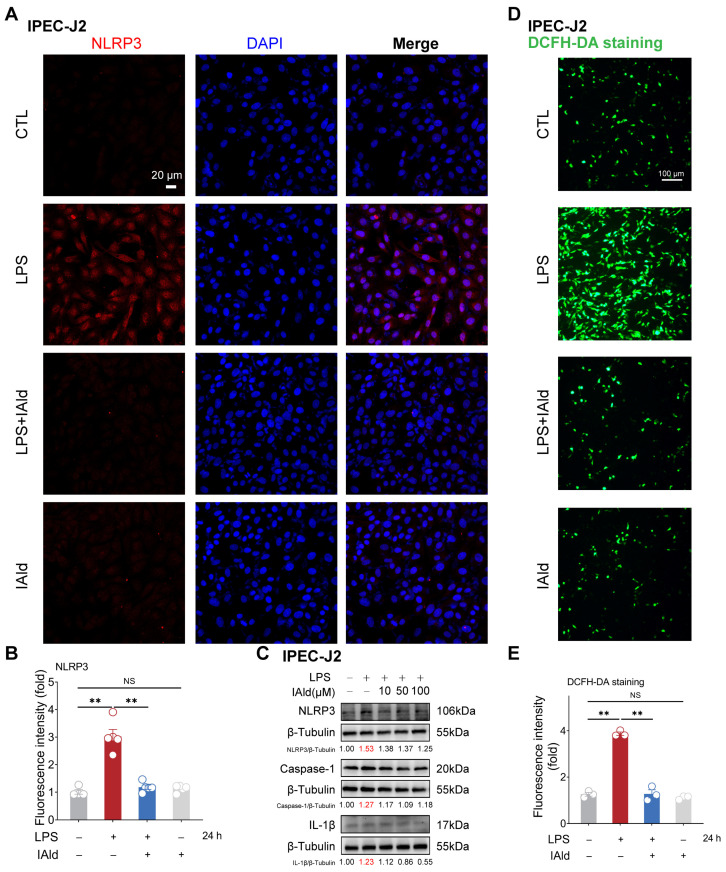
IAld inhibits LPS-induced activation of the NLRP3 inflammasome in IPEC-J2 cells. (**A**,**B**) IPEC-J2 cells were treated with IAld (0, 10 μM) in the presence or absence of LPS (10 μg/mL) stimulation for 24 h, the NLRP3 protein expression level was analyzed by immunofluorescence, scale bar = 20 μm. (**C**) IPEC-J2 cells were treated with different doses of IAld (0, 10, 50, and 100 μM) in the presence or absence of LPS stimulation, the NLRP3, caspase-1, and IL-1β protein expression levels were measured by Western blotting, and quantified by Image J software (Version 1.48). (**D**,**E**) After the indicated treatments, the intracellular ROS levels were detected by DCFH-DA staining, scale bar = 100 μm. ROS fluorescence intensity was quantified by Image J software (Version 1.48). Data are presented as means ± SEM (*n* = 3). ** *p* < 0.01, compared with the respective control. NS: no significance between the indicated groups.

**Figure 2 antioxidants-13-01107-f002:**
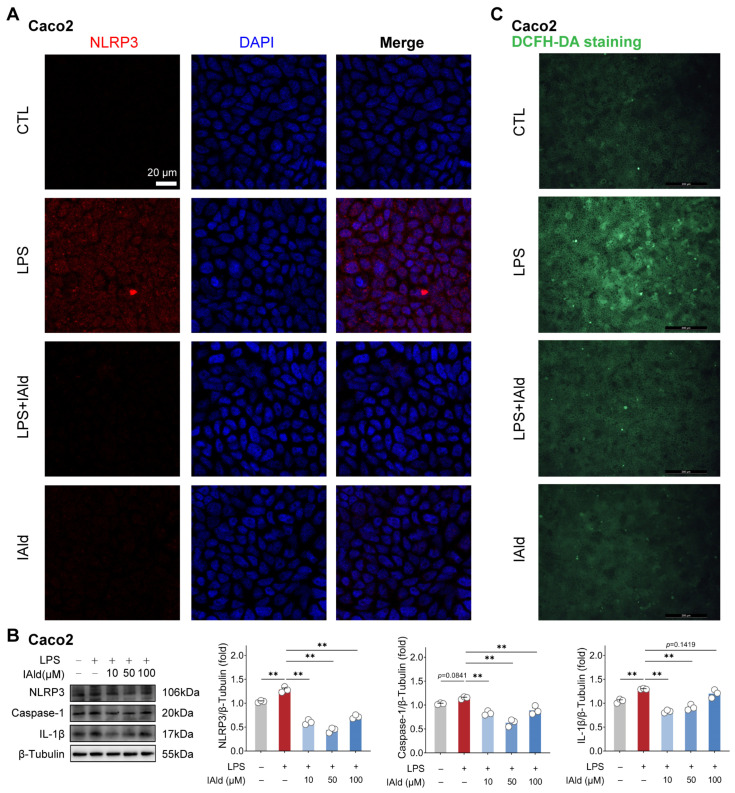
IAld inhibits LPS-induced activation of the NLRP3 inflammasome in Caco2 cells. (**A**) Caco2 cells were treated with IAld (0, 10 μM) in the presence or absence of LPS stimulation for 24 h, and the NLRP3 protein expression level was analyzed by immunofluorescence, scale bar = 20 μm. (**B**) Caco2 cells were treated with different doses of IAld (0, 10, 50, 100 μM) in the presence or absence of LPS stimulation, and the NLRP3 inflammasome-related protein (NLRP3, caspase-1, and IL-1β) expression levels were measured by Western blotting and quantified by Image J software (Version 1.48). (**C**) The intracellular ROS levels were detected by DCFH-DA staining, scale bar = 200 μm. Data are presented as means ± SEM (*n* = 3). ** *p* < 0.01, compared with the respective control. NS: no significance between the indicated groups.

**Figure 3 antioxidants-13-01107-f003:**
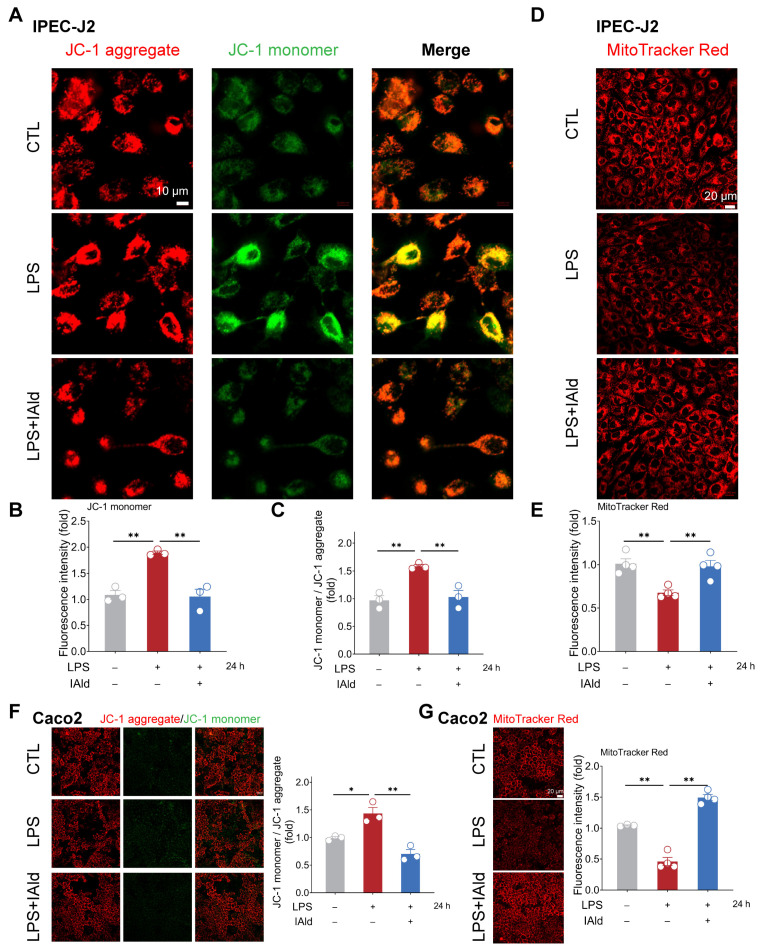
IAld prevents LPS-induced mitochondrial dysfunction in intestinal epithelial cells. (**A**–**C**) IPEC-J2 cells were treated with IAld in the presence or absence of LPS stimulation for 24 h, and the mitochondrial membrane potential levels were detected using JC-1 staining, scale bar = 10 μm. The fluorescence intensity of the JC-1 monomer and aggregate was quantified by Image J software (Version 1.48). (**D**,**E**) The mitochondrial membrane potential levels in IPEC-J2 cells were detected using MitoTracker Red staining, scale bar = 20 μm. The fluorescence intensity was quantified by Image J software (Version 1.48). (**F**) Caco2 cells were treated with IAld in the presence or absence of LPS stimulation for 24 h, and the mitochondrial membrane potential levels were detected using JC-1 staining and quantified by Image J software (Version 1.48), scale bar = 20 μm. (**G**) The mitochondrial membrane potential levels in Caco2 cells were detected using MitoTracker Red staining and quantified by Image J software (Version 1.48), scale bar = 20 μm. Data are presented as means ± SEM (*n* = 3). * *p* < 0.05, ** *p* < 0.01, compared with the respective control.

**Figure 4 antioxidants-13-01107-f004:**
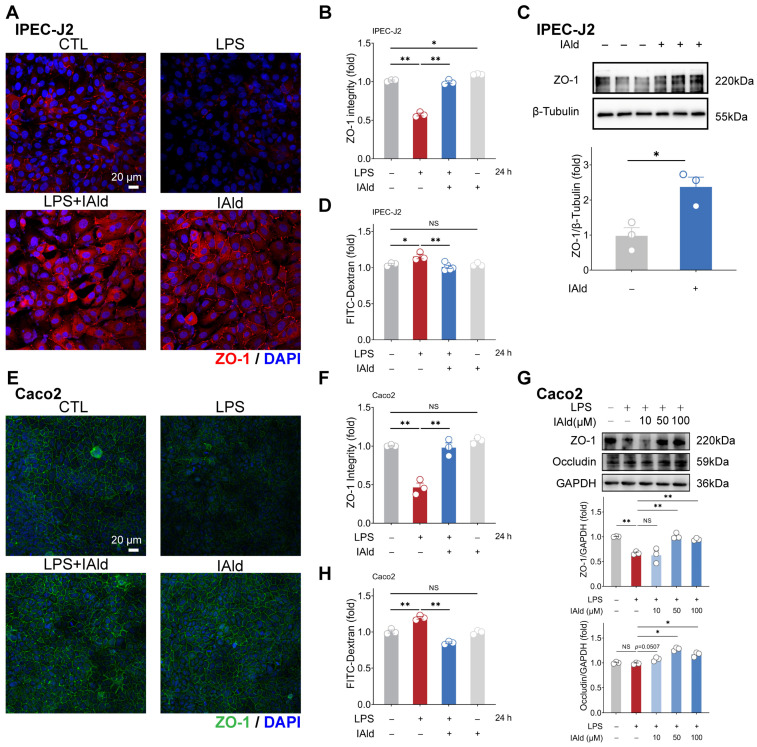
IAld alleviates LPS-induced barrier dysfunction in intestinal epithelial cells. (**A**,**B**) IPEC-J2 cells were treated with IAld in the presence or absence of LPS stimulation, and the distribution of tight junction protein ZO-1 was analyzed by immunofluorescence and quantified by Image J software (Version 1.48), scale bar = 20 μm. (**C**) After IAld (0, 10 μM) treatments, the ZO-1 expression levels were measured by Western blotting and quantified by Image J software (Version 1.48). (**D**) IPEC-J2 cells were treated with IAld in the presence or absence of LPS stimulation, and the FITC-Dextran (FD-4) leakage level was detected to reflect the permeability of intestinal epithelial cells. (**E**,**F**) Caco2 cells were treated with IAld in the presence or absence of LPS stimulation, and the distribution of tight junction protein ZO-1 was analyzed by immunofluorescence and quantified by Image J software (Version 1.48), scale bar = 20 μm. (**G**) Caco2 cells were treated with different doses of IAld (0, 10, 50, 100 μM) in the presence or absence of LPS stimulation, the tight junction protein (ZO-1, occludin) expression levels were measured by Western blotting and quantified by Image J software (Version 1.48). (**H**) Caco2 cells were treated with IAld in the presence or absence of LPS stimulation, and the FD-4 leakage level was detected. Data are presented as means ± SEM (*n* = 3). * *p* < 0.05, ** *p* < 0.01, compared with the respective control; NS: no significance between the indicated groups.

**Figure 5 antioxidants-13-01107-f005:**
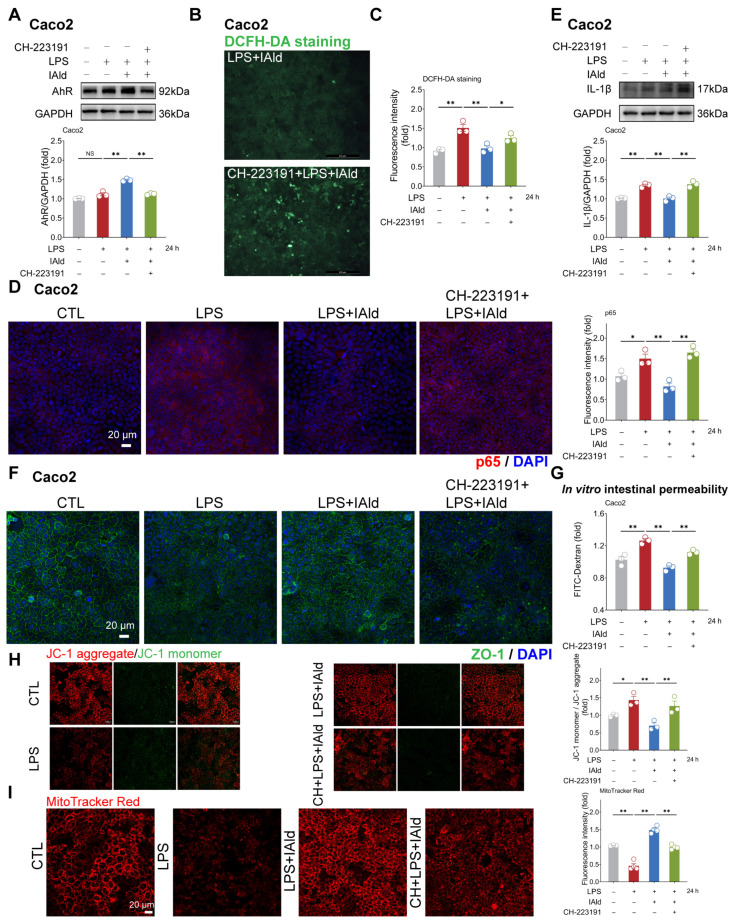
IAld inhibits intestinal inflammatory injury by activating the AhR in Caco2 cells. (**A**) Caco2 cells were treated with IAld in the presence or absence of AhR-specific inhibitor CH-223191 (10 μM), then stimulated with LPS for 24 h. The AhR protein expression level was measured by Western blotting and quantified by Image J software (Version 1.48). (**B**,**C**) After the indicated treatment, the intracellular ROS levels in Caco2 cells were detected by DCFH-DA staining, scale bar = 200 μm. The ROS fluorescence intensity was quantified by Image J software (Version 1.48). (**D**) The p65 protein expression level was analyzed by immunofluorescence, scale bar = 20 μm. The fluorescence intensity of p65 was quantified by Image J software (Version 1.48). (**E**) After the indicated treatment, the IL-1β protein expression level was measured by Western blotting and quantified by Image J software (Version 1.48). (**F**) After the indicated treatment, the distribution of tight junction protein ZO-1 was analyzed by immunofluorescence, scale bar = 20 μm. (**G**) After the indicated treatment, the FD-4 leakage level was detected to reflect the permeability of intestinal epithelial cells. (**H**,**I**) Mitochondrial membrane potential levels were detected by JC-1 and MitoTracker Red staining. Data are presented as means ± SEM (*n* = 3). * *p* < 0.05, ** *p* < 0.01, compared with the respective control; NS: no significance between the indicated groups.

**Figure 6 antioxidants-13-01107-f006:**
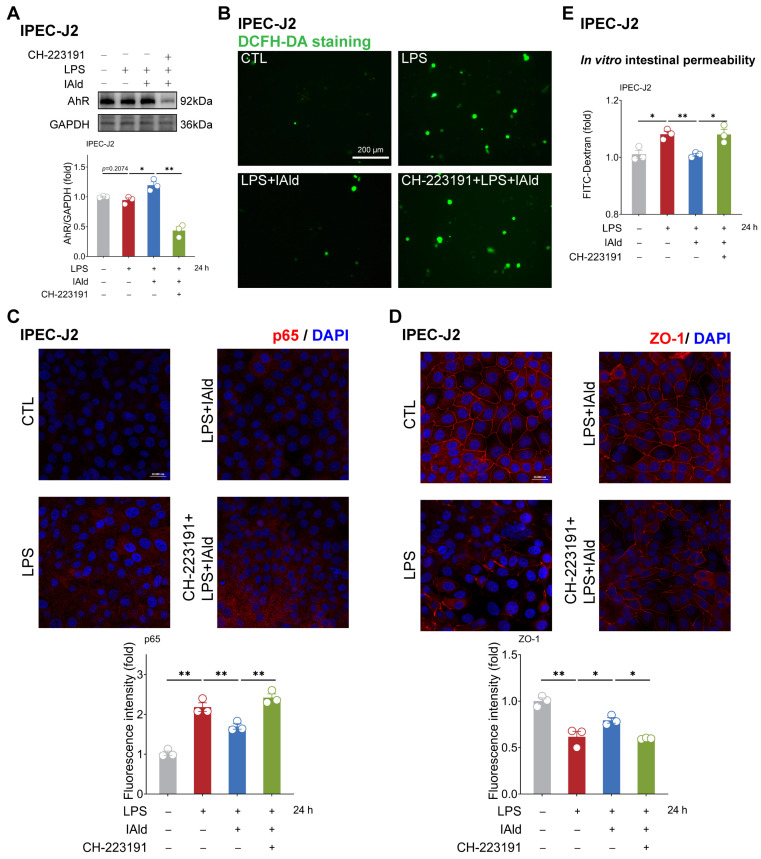
IAld inhibits intestinal inflammatory injury by activating the AhR in IPEC-J2 cells. (**A**) IPEC-J2 cells were treated with IAld in the presence or absence of AhR-specific inhibitor CH-223191, then stimulated with LPS for 24 h. The AhR protein expression level was measured by Western blotting and quantified by Image J software (Version 1.48). (**B**) After the indicated treatment, the intracellular ROS levels in IPEC-J2 cells were detected by DCFH-DA staining, scale bar = 200 μm. (**C**) The p65 protein expression level was analyzed by immunofluorescence and quantified by Image J software (Version 1.48), scale bar = 20 μm. (**D**) The tight junction protein ZO-1 distribution was analyzed by immunofluorescence and quantified by Image J software (Version 1.48), scale bar = 20 μm. (**E**) After the indicated treatment, the FD-4 leakage level was detected to reflect the permeability of intestinal epithelial cells. Data are presented as means ± SEM (*n* = 3). * *p* < 0.05, ** *p* < 0.01, compared with the respective control.

**Figure 7 antioxidants-13-01107-f007:**
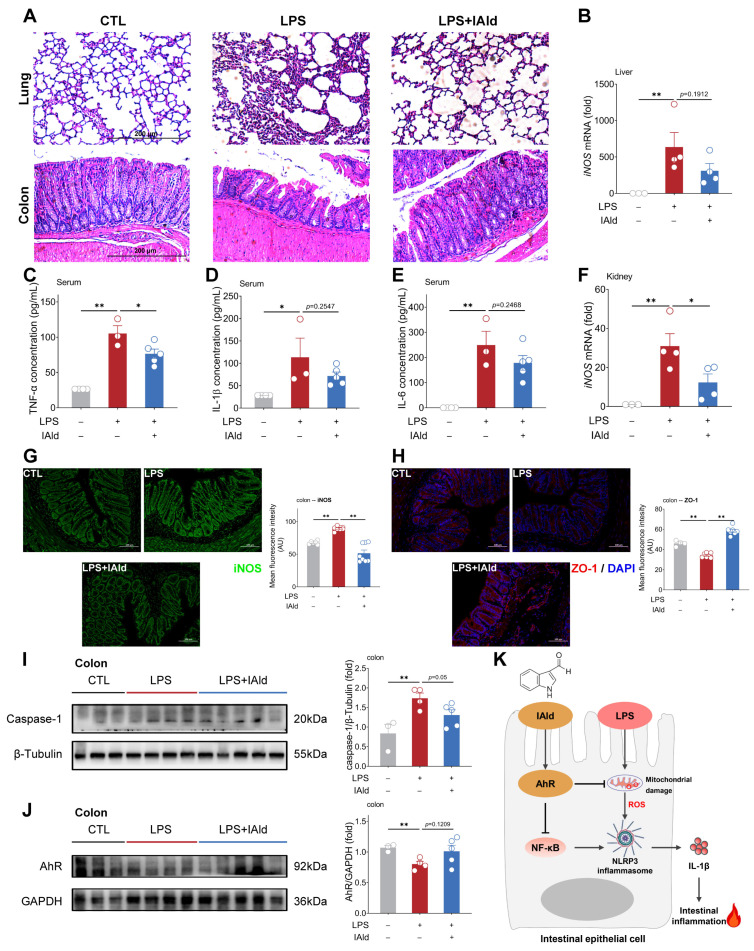
IAld alleviates LPS-induced intestinal inflammatory response in mice. (**A**) Representative images of H&E staining of the lung and colon from different treatment mice, scale bar = 200 μm. (**B**) The *iNOS* mRNA levels in the liver. (**C**–**E**) Serum TNF-α, IL-1β, and IL-6 concentrations. (**F**) The *iNOS* mRNA levels in the kidney. (**G**,**H**) The iNOS and ZO-1 protein expression levels in the colon were analyzed by immunofluorescence and quantified by Image J software (Version 1.48), scale bar = 100 μm. (**I**,**J**) The caspase-1 and AhR protein expression levels were measured by Western blotting and quantified by Image J software (Version 1.48). (**K**) Schematic diagram of the mechanism by which IAld alleviates LPS-induced inflammatory damage in intestinal epithelial cells. Data are presented as means ± SEM (*n* = 3 to 5). * *p* < 0.05, ** *p* < 0.01, compared with the respective control.

## Data Availability

All data are contained within the article. Detailed data can be obtained from the corresponding author.

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
