# Peer review of "Indole-3-Carboxaldehyde Alleviates LPS-Induced Intestinal Inflammation by Inhibiting ROS Production and NLRP3 Inflammasome Activation"

_antioxidants, 2024, doi:10.3390/antiox13091107_

Round 1

Reviewer 1 Report

In this study, the authors investigated the molecular mechanisms underlying the role of microbial-derived tryptophan (Trp) metabolite, IAld (Indole-3-carboxaldehyde) in the regulation of NLRP3 inflammasome activation and gut barrier function. The results showed that microbiota-derived IAld alleviated LPS-induced mitochondrial dysfunction, excessive production of ROS, and inflammatory damage in intestinal epithelial cells by activating AhR and inhibiting NLRP3 inflammasome both in vivo and in vitro. These findings further suggest AhR ligand, IAld as a potential endogenous NLRP3 inhibitor and effective antioxidant for treating intestinal inflammation. However, there are some concerns that the authors should consider:

 1. The rationale for using pig intestinal epithelial cells (IPEC-J2) is not clear.

2. Another indole derivative, indole 3 acetic acid has shown to ameliorate DSS colitis in mice via AhR/Nrf2/NLRP3 inflammasome pathway (Ref# 49). Did the authors investigate the role of the Nrf2 pathway in mediating the effects of IAld in attenuating intestinal inflammation?

3. Please mention the dose of LPS in cells and mice in the ‘Methods’ section

4. Figure 4A- The confocal images are not convincing. ZO1 staining in LPS appears to be more intense vs Control. The graph does not match the LPS confocal image.

5. Figure 7H- Intensity of ZO1 staining in both Control and LPS appear almost similar. Please provide a representative image that shows decreased ZO1 staining in LPS group compared to Control. Also include the densitometric analysis.

6.  Figures 5H and 5I- Please include the other 2 groups, Control and LPS alone

Please see the comments mentioned above

Reviewer 2 Report

The study is well done, informative and with many endpoints. A limit of the study is the missing information in the Materials and Methods section. Authors should mention that the in-vitro part could be very different from physiological condition, since the effects of LPS are different in the cellular systems vs whole organism, as reported by the literature.

Page 2, Lines 83-86:  You must remove this sentence. (”3. Results.......................... conclusions that can be drawn.”)

Regarding both in vitro and in vivo experiments, there are no indications in Materials and methods section on the LPS concentration. In the description of the in vivo experiments, there are no details on the number of mice used. Also, could the authors specify why an concentration of 50mg/kg b.w. IAld was chosen? 

For qPCR analysis, did the authors tested the % of efficacy of the primers used for the qPCR? Also, what are the analysed genes?

Description of the western blotting needs more details (e.g. the dilution of the primary and secondary Abs used).

4     Statistics: Did the authors check for the normal distribution of the data?

In the representation of qPCR results, in graphs, the expression of the interest genes was not reported to reference genes. To have reproductible qPCR results, the literature showed that there must to be used at least two reference genes, from a panel of 4-5 genes (Livak KJ, Schmittgen TD. Analysis of relative gene expression data using real-time quantitative PCR and the 2(-Delta Delta C(T)) Method. Methods. 2001 Dec;25(4):402-8. doi: 10.1006/meth.2001.1262. PMID: 11846609.). I recommend to use a group of reference genes, to select the best candidates using specific analysis methods (GeneNorm for example) and to re-analyse the qPCR data.

Reviewer 3 Report

This study investigated the protective effects of the gut microbiota-derived tryptophan metabolite indole-3-aldehyde (IAld) on lipopolysaccharide (LPS)-induced intestinal inflammation. The results showed that IAld inhibited the activation of the NLRP3 inflammasome in intestinal epithelial cells, prevented epithelial barrier damage, and alleviated inflammatory injury in an LPS-induced mouse model. Mechanistically, IAld activated the aryl hydrocarbon receptor (AhR), which subsequently suppressed the production of reactive oxygen species (ROS), maintained mitochondrial membrane potential, and blocked the NF-κB/NLRP3 inflammatory pathway. These findings suggest that IAld, as an effective endogenous antioxidant, could be a potential therapeutic target for NLRP3-induced inflammatory bowel diseases.

The article lacks detailed descriptions of experimental methods, such as cell culture conditions and animal experiment design, which may affect the reproducibility and reliability of the results. It is recommended to supplement the experimental section with more detailed information.

Some of the result images in the article are of poor quality, such as Figures 6C and 6D, which may hinder the readers' understanding of the results. It is suggested to improve the image quality to ensure clarity and readability.

The discussion section is too simplistic, lacking in-depth analysis and interpretation of the experimental results. It is recommended to enhance the discussion by providing more detailed explanations of the findings and comparing them with existing studies, in order to better elucidate the mechanism of IAld regulation of the NLRP3 inflammasome.

The conclusion section is overly brief, lacking an explanation of the significance of the study and future research directions. It is suggested to include a discussion of the importance of IAld as a potential therapeutic target and the research directions that need further exploration.

The article lacks a discussion of the study’s limitations, such as the limitations of the experimental models and sample sizes. It is recommended to add an analysis of the limitations in the discussion section to provide a reference for future research.

Reviewer 4 Report

Author used porcine and human cell lines to evaluate IAld’s effectiveness on mitigating inflammatory responds induced by LPS O55B5. There are some concerns regarding to the variation between different cell type and in vivo experiment and presentation of results. 

Line 75: was the LPS O55:B5 having pathogenic effect on both pigs and human? Why not used mice intestinal cell line for vitro to avoid confounding effect by species?

Line 99: was 50 mg/kg the dosage for LPS or IAld?

Can IAld remained its integrity through stomach digestion?   

Line 142: Please specify the model and experimental unit used for each analysis.  Were data normality determined for each set of data?

Figure 1B please make sure that significant labels are correct.

Figure 2: the resolution of imagines in figure 2 are poor.

Figure 2c: why did increased IAld result in increasing IL-1β concentration?

It looks like the species difference does exist on the responses of LPS O55:B5.

The resolution on Figure 3F is poor to where LPS didn’t show much impact by color density. Please provide the bar chart results as figure 3c. 

Figure 4D: How was the result of FITC-dextran when treated with IAld alone?

 Figure 5: how was the result of IAld alone on these traits?

Line 293: It is hard to differentiate the color intensity between LPS+IAld and LPS+IAld+CH on figure 6D.

Figure 6 CD: Please provide bar charts the same as figure 5.

Line 311 to 322: These statements are in contrast with results shown in Figure 7.  

Round 2

Reviewer 1 Report

The authors have duly addressed all the concerns raised by the reviewer

Please see as mentioned above

Reviewer 2 Report

The authors revised the entire paper, and the present form is suitable for publication.

The authors revised the entire paper, and the present form is suitable for publication.

Reviewer 4 Report

The author has addressed my comments

The author has addressed my comments